

# Molecular genetic diversity of seaweeds morphologically related to *Ulva rigida* at three sites along the French Atlantic coast

Manon Dartois[1], Eric Pante[1,2], Amélia Viricel[1], Vanessa Becquet[1] and Pierre-Guy Sauriau[1]

[1] Littoral, Environnement et Sociétés, UMR 7266, CNRS - La Rochelle Université, La Rochelle, France
[2] Institut Systématique Evolution Biodiversité (ISYEB), CNRS, Sorbonne Université, EPHE, Université des Antilles, Museum national d'Histoire naturelle, Paris, France

Corresponding authors
Manon Dartois,
manon.dartois0@gmail.com
Eric Pante, epante@univ-lr.fr

## ABSTRACT

Foliose species of the genus *Ulva* are notoriously difficult to identify due to their variable morphological characteristics and high phenotypic plasticity. We reassessed the taxonomic status of several distromatic foliose *Ulva* spp., morphologically related to *Ulva rigida,* using DNA barcoding with the chloroplastic *tuf*A and *rbc*L (for a subset of taxa) genes for 339 selected attached *Ulva* specimens collected from three intertidal rocky sites. Two of the collection sites were in Brittany and one site was in Vendée, along the Atlantic coast of France. Molecular analyses included several museum specimens and the holotype of *Ulva armoricana* Dion, Reviers & Coat. We identified five different *tuf*A haplotypes using a combination of phylogenetic analysis, with the support of several recently sequenced holotypes and lectotypes, and a species delimitation method based on hierarchical clustering. Four haplotypes were supported by validly named species: *Ulva australis* Areschoug, *Ulva fenestrata* Postels & Ruprecht, *Ulva lacinulata* (Kützing) Wittrock and *U. rigida* C. Agardh. The later was additionally investigated using *rbc*L. The fifth haplotype represented exact sequence matches to an unnamed species from European Atlantic coasts. Our results support: (1) the synonymy of both *U. rigida sensu* Bliding *non* C. Agardh and *U. armoricana* with *U. lacinulata*. This finding is based on current genetic analysis of *tuf*A from the *U. armoricana* holotype and recent molecular characterization of the lectotype of *U. laetevirens,* which is synonymous to *U. australis*, (2) the presence of *U. australis* as a misidentified introduced species in Brittany, and (3) the presence of *U. fenestrata* and *U. rigida* in southern Brittany. The taxonomic history of each species is discussed, highlighting issues within distromatic foliose taxa of the genus *Ulva* and the need to genetically characterize all its available type specimens.

## INTRODUCTION

Macroalgae proliferations in coastal environments fuelled by anthropogenic eutrophication (*Fletcher, 1996*; *Ye et al., 2011*) are a worldwide phenomenon (*Smetacek & Zingone, 2013*;
*Wan et al., 2017*). Most are composed of species in the genus *Ulva* (*Fletcher, 1996*; *Jia et al., 2011*), leading to the aptly-named 'green tides.' These are composed of free-floating thalli that may become stranded on sheltered areas. Environmental changes affect both pelagic and benthic communities and are detrimental to the ecology, economy, and sanitation of coastal areas (*Charlier, Morand & Finkl, 2008*; *Ye et al., 2011*; *Smetacek & Zingone, 2013*). Huge algal biomasses increase sedimentation rates and interfere with oxygen transport. Algae consume oxygen during respiration and create anoxic conditions, followed by the decomposition of algal mats and the development of toxic gaseous sulphur compounds within the stranded biomass (*Fletcher, 1996*; *Charlier, Morand & Finkl, 2008*). Human poisoning and deaths have even been reported following inhalation of hydrogen sulphide (*Ménesguen, 2018*).

One of the main challenges in green tide studies is to characterize the *Ulva* species involved. Identifying the species can answer key biological questions, including the level of pluri-specificity (*Coat et al., 1998*; *Malta, Draisma & Kamermans, 1999*; *Kang et al., 2014*; *Fort et al., 2020*), occurrence of undescribed species (*Dion, De Reviers & Coat, 1998*; *Lee, Kang & Kim, 2019*), allochthonous/exogeneous specific status (*Wolf et al., 2012*; *Steinhagen, Karez & Weinberger, 2019*), biological mechanisms underlying algal growth (*De Casabianca et al., 2002*; *Fort et al., 2019*), and differences between free-floating and attached thalli (*Malta, Draisma & Kamermans, 1999*; *Han et al., 2013*; *Zhao et al., 2015*). Efforts have been made to describe the *Ulva* species, provide synopses of reliable morphological and anatomical characteristics, and disentangle taxonomic confusions (*Bliding, 1969*; *Koeman & and Van den Hoek, 1981*; *Hoeksema & and Van den Hoek, 1983*; *Phillips, 1988*), but misidentification and taxonomic confusion are still common, particularly amongst foliose *Ulva* species (*Loughnane et al., 2008*; *Kraft, Kraft & Waller, 2010*; *Kirkendale, Saunders & Winberg, 2013*; *Hughey et al., 2021b*; *Fort et al., 2021b*). In some cases, this confusion has led to the coexistence of divergent interpretations of taxa with the same specific epithet, for example, *Ulva rigida* C. Agardh and *Ulva rigida sensu* Bliding *non* C. Agardh. The later species was referred to as *Ulva laetevirens* Areschoug according to *Phillips (1988)*, and this view was endorsed by numerous studies (*Kraft, Kraft & Waller, 2010*; *Sfriso, 2010*; *Cormaci, Furnari & Alongi, 2014*; *Mao et al., 2014*). This opinion, however, was not widely accepted (*Womersley, 1984*) as *Gallardo et al. (1993)*, *Verlaque, Belsher & Deslous-Paoli (2002)* and *Loughnane et al. (2008)* all argued for further morphological investigation, particularly on type material. New species related to *U. rigida sensu* Bliding such as *Ulva scandinavica Bliding (1969)* and *Ulva armoricana* were described in Europe (*Dion, De Reviers & Coat, 1998*).

The use of morphological characteristics alone to identify species in the *Ulva* genus is often insufficient due to phenotypic plasticity within the genus and the role of associated bacteria on macroalgal morphogenesis (*Alsufyani et al., 2020*). Molecular analyses are used in species delineation and phylogenetic studies as alternatives to morphology (*Hayden & Waaland, 2004*; *Loughnane et al., 2008*; *Kraft, Kraft & Waller, 2010*), but even these methods are useless unless used in a rigorous taxonomic framework. It has been argued that, based on their morphological and cytological characteristics, the species responsible for local green tides in Brittany during the 1990s include *Ulva rotundata* Bliding and *U. armoricana*

(*Dion, De Reviers & Coat, 1998*). *Coat et al. (1998)* used molecular analysis to highlight similarities in ITS rDNA sequences between *U. rotundata* from Brittany and material labelled *U. rigida* from Australia. This unanticipated similarity was confirmed by *Malta, Draisma & Kamermans (1999)* and was further investigated by *Shimada et al. (2003)*, *Hayden et al. (2003)*, *Hayden & Waaland (2004)*, and *Couceiro, Cremades & Barreiro (2011)*, who finally established the conspecificity between '*U. rotundata*' specimens from Brittany and *U. australis* Areschoug from Australia. In addition, *U. armoricana* may be conspecific with *U. 'rigida'* based on ITS (*Malta, Draisma & Kamermans, 1999*; *Hayden et al., 2003*; *Shimada et al., 2003*; *O'Kelly et al., 2010*), ITS combined with *rbc*L (*Hayden & Waaland, 2004*), and *rbc*L alone (*Loughnane et al., 2008*). *Loughnane et al. (2008)* and *Miladi et al. (2018)* also suggested that *U. rigida* C. Agardh and *U. laetevirens* Areschoug respective specific statuses still require morphological and molecular analyses of type materials to be distinguished. *Hughey et al. (2021a)* and *Hughey et al. (2021b)* provide a convincing answer to the questionable relatedness of *U. laetevirens* with *U. australis* using *rbc*L sequencing. On one hand, they established that *U. laetevirens* is a heterotypic synonym of *U. australis*, based on lectotypes of both taxa (*Hughey et al., 2021a*). On the other hand, *Hughey et al. (2021b)* argued that all published sequences of *U. laetevirens* (= *U. rigida sensu* Bliding) in gene repositories are erroneously named and should be assigned to *U. lacinulata* (Kützing) Wittrock. Taxonomic reappraisals can even contribute to the current difficulties in synonymising *U. armoricana* and *U. scandinavica*. Conspecificity with *U. rigida* C. Agardh was the previously accepted view (*Brodie, Maggs & John, 2007*), although most molecular studies addressing this hypothesis referred to *Bliding (1969)* and Phillips' (1988) morphological categorization of *U. rigida* as *U. rigida sensu* Bliding (*Hayden et al., 2003*; *Shimada et al., 2003*; *Loughnane et al., 2008*; *Kraft, Kraft & Waller, 2010*; *Kirkendale, Saunders & Winberg, 2013*; *Mao et al., 2014*; *Wan et al., 2017*). Conspecificity of *U. scandinavica* with *U. rigida sensu* Bliding (= *U. laetevirens*) was promoted by *Kraft, Kraft & Waller (2010)* and *Kirkendale, Saunders & Winberg (2013)* on the basis of molecular analyses.

Molecular analyses, practices, and protocols in DNA-based species identification have been strengthened in several ways: (1) *Saunders & Kucera (2010)* recommended the plastid elongation factor *tuf*A instead of ITS rDNA and plastid gene *rbc*L in barcoding green marine macroalgae, (2) large sample sizes guarantee better analytical robustness and intraspecific variability estimates at the population level, and (3) the use of museum-type specimens allow tests of species hypotheses to be unequivocal (*Pante et al., 2015*; *Hughey et al., 2019*; *Hughey et al., 2021a*; *Hughey et al., 2021b*). In fact, the chloroplastic elongation factor *tuf*A marker has been developed for routine barcoding of green marine macroalgae, excluding the Cladophoraceae (*Saunders & Kucera, 2010*). Previous studies on *Ulva* spp. using *tuf*A suggest that it is variable enough to allow the comparison of intra- and interspecific variation across *Ulva* species, making it a useful molecular barcode for the genus (*Kirkendale, Saunders & Winberg, 2013*; *Kang et al., 2014*; *Miladi et al., 2018*; *Lee, Kang & Kim, 2019*; *Steinhagen, Karez & Weinberger, 2019*). The use of several different genetic markers within the *Ulva* genome (either mitochondrial, chloroplastic or nuclear) nevertheless adds an unexpected difficulty when comparing results and identifying species.

For example, *U. rotundata*, which was synonymised as *U. pseudorotundata* Cormaci, G. Furnari & Alongi, has been identified based on *rbc*L sequencing (*Loughnane et al., 2008*; *Wan et al., 2017*) and not on *tuf*A, except for a unique study (*Fort et al., 2021a*). However, analysis of the *rbc*L sequence of the holotype of *U. rotundata* supports the conclusion that *U. rotundata* is a heterotypic synonym of *Ulva lactuca* Linnaeus (*Hughey et al., 2021b*). The use of different primers, sequence lengths, and/or the addition of new available specific sequences may result in slight discrepancies between studies. It is worth noting that results based on large sample sizes and datasets (*Couceiro, Cremades & Barreiro, 2011*; *Kirkendale, Saunders & Winberg, 2013*; *Hanyuda et al., 2016*; *Lee, Kang & Kim, 2019*; *Steinhagen, Karez & Weinberger, 2019*), and museum-type material (*Hanyuda & Kawai, 2018*; *Hughey et al., 2019*; *Hughey et al., 2021a*; *Hughey et al., 2021b*; *Steinhagen, Karez & Weinberger, 2019*) have contributed significantly to clarifying *Ulva* spp. taxonomy. The development of organellar (chloroplast and mitochondrion) genome sequencing, combined with species delimitation models, also represent a major step towards a more comprehensive estimate of intra- and interspecific genetic variability (*Fort et al., 2021a*; *Fort et al., 2021b*).

We sought to reassess the genetic diversity of foliose *Ulva* species morphologically related to *Ulva rigida* sampled from several sites along the French Atlantic coasts. Our approach consisted of a phylogenetic analysis of *tuf*A combined with the chloroplast-encoded *rbc*L gene for a subset of taxa. We also included a large sample size, the type locality of *U. armoricana* in Brittany, and analyses of museum material. We sampled 360 thalli with the macro-morphological characteristics of foliose *U.* cf. *rigida* from the intertidal rocky shores of two sites in Brittany and one site in Vendée in the winter. We collected only attached thalli to avoid stranded material as these sites suffer from summer to autumn green tides of free-floating thalli (*CEVA, 2015*, *CEVA, 2019*; *Merceron & Morand, 2004*). Our study is the first to include such large numbers of *Ulva* samples from several sites on the French Atlantic coasts, compared to historical (*Coat et al., 1998*) or more recent (*Fort et al., 2020*; *Fort et al., 2021a*; *Fort et al., 2021b*) molecular studies. We also analysed the *tuf*A sequence of the *U. armoricana* holotype collected by *Dion, De Reviers & Coat (1998)* at Roscoff (Museum national d'Histoire naturelle, MNHN, Paris, France; voucher MNHN-PC-PC0115137) to clarify the taxonomic relationships between *U. armoricana* and other *Ulva* species related to *U. rigida*, and confirm synonymies, particularly in view of genetic analyses of the lectotype specimens of *U. rigida* and *U. lacinulata* recently provided by *Hughey et al. (2021b)*.

## MATERIAL AND METHODS

### Sampling

Sampling was performed between January 22th and February 21st 2019 in the intertidal zone of three sites: La Tranche sur Mer (46° 20′48.6″N 1°25′19.3″W) in Vendée, Roscoff (48°43′48.1″N 3°58′57.7″W), and Concarneau - Cabellou (47°51′34.6″N 3°54′47.9″W) in Brittany (Fig. 1). We collected samples during the January/February period to avoid the proliferation of seasonal *Ulva* population known to occur at these sites (*CEVA, 2015*, *CEVA, 2019*;), during which mostly haploid individuals are produced (*Potter et al., 2016*). This allowed us to capture diploid individuals for further species delimitation

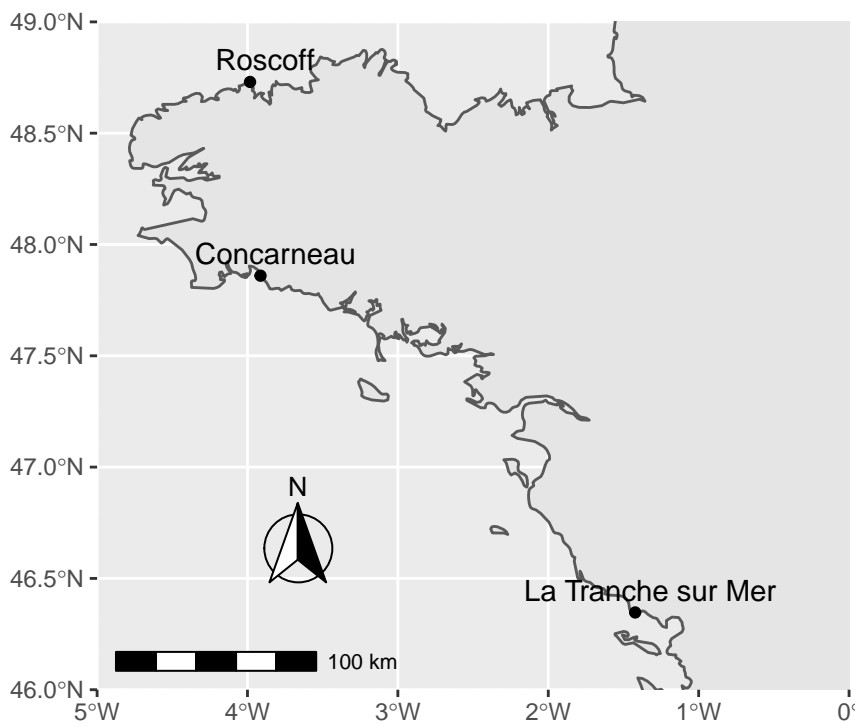

**Figure 1** Map of sampling sites along the French Atlantic coast (Roscoff and Concarneau in Brittany, and La Tranche sur Mer in Vendée).

using nuclear RAD loci (this project is ongoing). Only attached and whole foliose thalli greater than 7 cm² were collected from rocky substrates extending over the intertidal zone. Any free-floating thalli from remote intertidal sites or subtidal locations were discarded (*Merceron & Morand, 2004*). Field identification was based on green foliose macroalgae matching the macro-morphology of *Ulva rigida* with large and flat thallus, a bright green colour, and stiff base (*Phillips, 1988*; *Loiseaux-de Goër & Noailles, 2008*; *Loughnane et al., 2008*; *Sfriso, 2010*). At each site, more than 200 specimens were collected into individual plastic bags and kept at 4 °C. Each sample was rinsed with filtered seawater in the lab to remove epiphytes and was checked for the presence of stiff basal and rhizoidal regions (*Sfriso, 2010*), the absence of sporulation or gametogenesis in thallus margins, and the presence of a distromatic blade (observed in transverse sections under a light microscope). We did not consider other cellular criteria (*Bliding, 1969*); *Koeman & Van den Hoeck, 1981*; *Hoeksema & Van den Hoek, 1983*), taking into account their natural variability within and between foliose distromatic *Ulva* species (*Coat et al., 1998*; *Loughnane et al., 2008*; *Kraft, Kraft & Waller, 2010*). Approximately 120 specimens per site were collected and preserved at −80 °C in individually-numbered plastic bags. Eleven museum samples from the cryptogam collection (PC) of the Muséum national d'Histoire naturelle, Paris (France), including the holotype of *U. armoricana* (*Dion, De Reviers & Coat, 1998*), were added to our samples (Supplemental S1).

**Table 1** Parameters and sequences of *tuf*A primers, based on Saunders & Kucera (2010).

| Primer name | Tm | Sequence (5′–3′) | Expected amplicon length (bp) |
|---|---|---|---|
| *tuf*GF4_MD (Forward) | 58.5 °C | GGTGCAGCYCAAATGGATGG | 800 |
| *tuf*AR_MD (Reverse) | 63.3 °C | CCTTCACGAATTGCAAAACGC | |

**Notes.**

Tm, Melting temperature; bp, base-pair.

## DNA extraction and PCR amplifications

Frozen tissue from the thallus was ground to a powder in liquid nitrogen. Whole genomic DNA was extracted from 0.3 mg samples of the powder using the NucleoSpin Tissue Kit (Macherey-Nagel). The manufacturer's standard protocols for tissues were followed, except for the following steps: (1) we performed an overnight tissue digestion in proteinase K, (2) DNA was eluted in two steps, each with a 3 min incubation with 25 μL of dH$_2$O pre-heated at 70 °C, for a final volume of 50 μL. DNA quality and quantity were assessed using a Nanodrop ND-2000 spectrophotometer (Thermo Scientific), a Qubit 1.0 (Thermo Scientific) fluorometer (dsDNA HS Assay Kit), and 1X agarose gel electrophoresis.

The chloroplast gene *tuf*A was targeted to barcode our specimens. Primers were designed based on *Saunders & Kucera (2010)* to reduce the number of ambiguities based on the chloroplast genomes available for *Ulva* in GenBank (*Ulva* sp. KP720616.1, *Ulva flexuosa* KX579943.1, NC_035823.1, *Ulva prolifera* NC_036137.1, KX342867.1, *Ulva ohnoi* AP018696.1, *Ulva lactuca* NC_042255.1, MH730972.1, *Ulva linza* KX058323.1, NC_030312.1 and *Ulva fasciata* NC_029040.1, KT882614.1). Primer sequences are shown in Table 1. PCR was carried out using a Sensoquest labcycler with a TaKaRa ExTaq reaction kit (Takara Bio). PCR amplicons were checked on a 1X agarose gel electrophoresis prior to purification and Sanger sequencing in both forward and reverse directions by Eurofins Genomics (Ebersberg, Germany). Negative controls were performed at the extraction and PCR amplification steps.

All *tuf*A sequences (plus one *rbc*L sequence), including sequences from MNHN specimens, were deposited in GenBank (Supplemental S1 and S7).

## Data analysis

Chromatograms were cleaned manually with Geneious Prime 2019.1.2 (http://www.geneious.com/), primer sequences were trimmed, sequences were checked for ambiguities and stop codons. Forward and reverse sequences were then assembled. The final sequence length of the *Ulva* specimens varied between 807 and 877 bp (Supplemental S1). All *tuf*A sequences produced in this study were aligned to 1,517 available *Ulva* spp. sequences from GenBank using Muscle 3.8.425 (*Edgar, 2004*). Three *Umbraulva japonica* sequences, 14 *Umbraulva* sp. and one *Umbraulva dangeardii* were added to constitute an outgroup. Identical sequences from the same species were represented by a single *tuf*A haplotype for further phylogenetic analyses but, when available, holotype or lectotype specimens were highlighted. This resulted in the selection of 139 and four *Ulva* and *Umbraulva* haplotypes, respectively. Uncorrected p distances (hereafter called p distances) were calculated using PAUP* v.4.0 (*Swofford, 2002*) based on the *tuf*A sequences of 774 bp (available on GenBank

for *Kirkendale, Saunders & Winberg, 2013*), and truncated sequences of 500 bp, so as to allow alignment with other *Ulva* sequences available on GenBank. Truncating the alignment to 500 bp does not change the number of haplotypes within our data set and allows the inclusion of more *Ulva* species sequences from GenBank. The full tree with 143 sequences was reduced for clarity due to the large number of species included. We excluded species such as *Ulva compressa*, *U. flexuosa*, *Ulva intestinalis*, *U. linza*, *U. prolifera*, *Ulva stenophylla* and *Ulva torta* to be consistent with our analysis of Brittany and Vendée foliose specimens (*Hoeksema & and Van den Hoek, 1983*; *Loiseaux-de Goër & Noailles, 2008*). We also excluded all *Ulva* sp. not related to our results and reduced the outgroup to only *Umbraulva japonica*. Our reduced tree (Fig. 2) was based on 1,185 sequences (80 *Ulva* sequences and one *Umbraulva* haplotype as the outgroup) and supported the results from the full dataset (Supplemental S2). Maximum Likelihood *tuf*A trees were inferred for *Ulva* species using IQ-TREE 2.0.5 (*Nguyen et al., 2015*; *Minh et al., 2020*) with ultrafast bootstrapping (1,000 pseudoreplicates) (*Hoang et al., 2018*) and a TPM3+F+I+G4 model of evolution (model selection performed using ModelFinder; *Kalyaanamoorthy et al., 2017*). The resulting trees were edited in Inkscape (https://inkscape.org/).

Species delimitation was performed using Assemble Species by Automatic Partitioning (ASAP; *Puillandre, Brouillet & Achaz, 2020*) and the Kimura-2-Parameter model of nucleotide substitution (*Kimura, 1980*), which was the closest model to the most likely model selected with ModelFinder. ASAP performs hierarchical clustering on genetic distances to split datasets into species partitions; partitions are attributed a robustness or the asap-score, based on the averaged ranked partition *p*-values and relative barcode gap width (*Puillandre, Brouillet & Achaz, 2020*). ASAP analyses were performed online (https://bioinfo.mnhn.fr/abi/public/asap/) on the 500 bp alignment previously described (80 *Ulva* haplotypes), and a 774 bp alignment (41 *Ulva* haplotypes) corresponding to the *tuf*A fragment described by *Kirkendale, Saunders & Winberg (2013)*. A complementary ASAP analysis was also performed on all available *Ulva* sp. complete *tuf*A sequences (1,224 bp) from GenBank, which returned 136 non-duplicated records distributed as follows: *U. australis* (17), *U. compressa* (7), *Ulva expansa* (2), *Ulva fasciata* (1), *U. fenestrata* (12), *U. flexuosa* (1), *Ulva gigantea* (10), *U. intestinalis* (1), *U. lacinulata* (38), *U. lacinulata* lectotype from Hvar (Lessina) in Croatia (1), *U. lactuca* (1), *U. linza* (1), *Ulva mutabilis* (1), *U. ohnoi* (1), *Ulva pertusa* (1), *U. prolifera* (1), *U. rigida* (7), *U. rigida* lectotype from Cadiz (1), *U. rotundata* (1), *U.* sp. A AF-2021 (29) as reported by *Fort et al. (2021b)* and *U.* sp. (2).

Haplotype richness ($R$), Shannon's diversity index ($H$) and Pielou's evenness ($J$) were calculated using vegan 2.5-4 (*Oksanen et al., 2019*) in R 3.6.0 (*R Core Team, 2020*). We used the same package to perform species rarefaction based on sample numbers and fit a Preston's veil model (method: maximized likelihood to log2 abundances) to our data (sites were pooled, Supplemental S3 and Supplemental S4) (*Preston, 1948*; *Williamson & Gaston, 2005*). Sampling sites were mapped using R 3.6.0 (Supplemental S6).

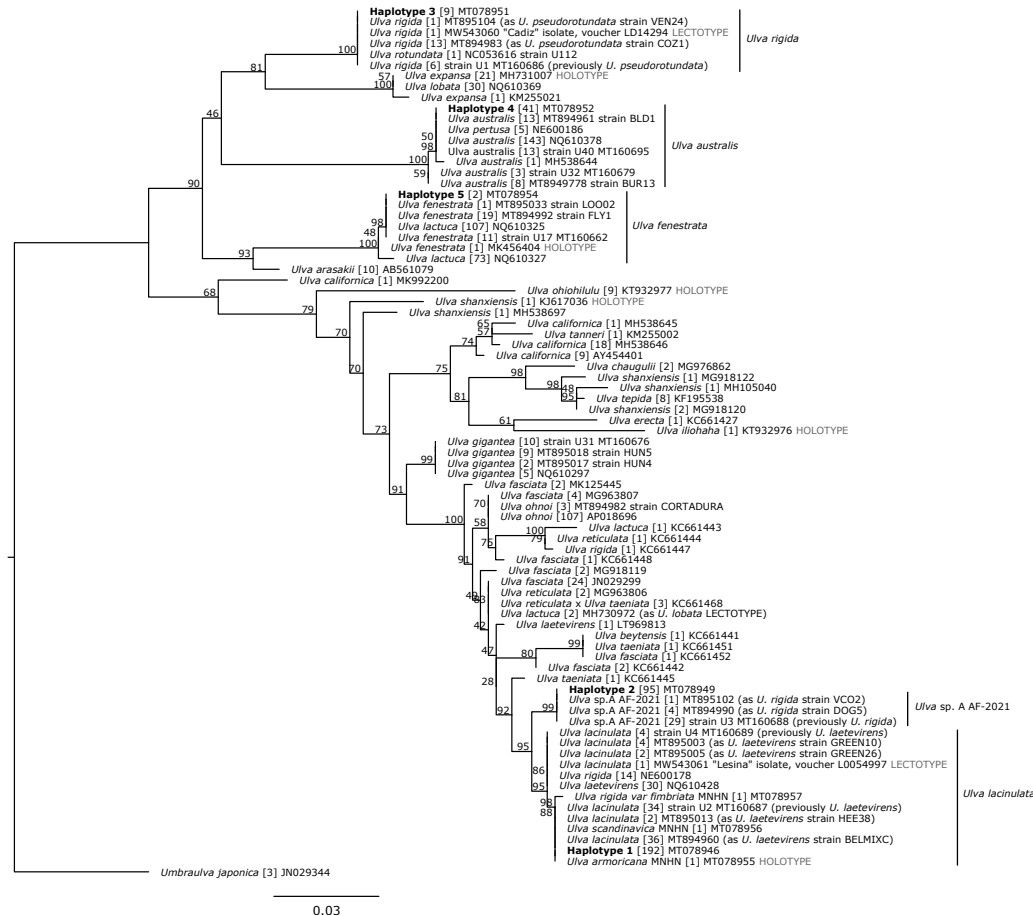

**Figure 2** **Maximum Likelihood (ML) phylogeny based on 500 bp of the *tuf*A chloroplastic gene.** Haplotypes detected in this study are in bold. Bootstrap support values from the ML analysis are indicated under each node. Sample size, for each haplotype, is presented in brackets. Unit of scale bar: substitution/site. MNHN: Muséum National d'Histoire Naturelle, Paris. Taxon names follow Genbank records, as of the time of publication.

## RESULTS

### *tuf*A analysis

*tuf*A was sequenced for 339 of the 360 samples and for three of the 11 MNHN specimens, because of amplification or sequencing difficulties (*U. armoricana* MNHN-PC-PC0115137, *Ulva rigida* var. *fimbriata* J. Agardh MNHN-PC-PC0531492 and *U. scandinavica* MNHN-PC-PC0547277). Five haplotypes were detected based on both the 500 and the 774 bp-long sequence alignments. Haplotype 1 was sampled at all sites. Haplotypes 2 and 4 were sampled in Brittany only (Concarneau and Roscoff), while haplotypes 3 and 5 were solely found at Concarneau, and sampled in small numbers *i.e.,* <10 thalli (Tables 2 and 3). Concarneau had the highest haplotypic richness ($R = 5$) and diversity ($H = 1.178$), followed by Roscoff ($R = 3$, $H = 1.095$) and La Tranche ($R = 1$), where only the most common haplotype (haplotype 1) was found. The haplotype distribution was even greater

**Table 2 Number of samples per haplotype at each site.**

|  | La Tranche s/Mer | Concarneau | Roscoff |
|---|---|---|---|
| Haplotype 1 | 118 | 36 | 38 |
| Haplotype 2 | 0 | 61 | 34 |
| Haplotype 3 | 0 | 9 | 0 |
| Haplotype 4 | 0 | 10 | 31 |
| Haplotype 5 | 0 | 2 | 0 |

**Table 3 Percent p distances for each pair of *tuf*A haplotypes, for the 500 bp (left value) and 774 bp (right value, in parentheses) alignment lengths.**

|  | Haplotype 2 | Haplotype 3 | Haplotype 4 | Haplotype 5 |
|---|---|---|---|---|
| Haplotype 1 | 1.2 (0.9) | 8.2 (7.6) | 10.2 (9.3) | 9.6 (8.4) |
| Haplotype 2 |  | 8.4 (7.9) | 10.4 (9.6) | 9.8 (8.6) |
| Haplotype 3 |  |  | 6.8 (6.5) | 6.6 (5.2) |
| Haplotype 4 |  |  |  | 7.0 (5.7) |

in Roscoff ($J = 0.9968$) than Concarneau ($J = 0.7318$). Rarefaction suggests that haplotype diversity was accurately estimated, as the rarefaction curve almost reaches an asymptote (Supplemental S3). Preston's Lognormal Model to Abundance Data suggested that 0.05 haplotypes were missed during sampling (5.0538 haplotypes were extrapolated with the method).

The 500 bp alignment based on 80 *Ulva* haplotypes contained 135 variable sites and 99 parsimony-informative sites. No indel was detected (alignment provided as Supplemental S8). On the ML tree, the five haplotypes were aligned with sequences of nominal *Ulva* species, including available holotype and lectotype sequenced specimens, and *Umbraulva japonica* (Fig. 2, Supplemental S5). To help evaluate the number of nominal species on the basis of genetic divergence, raw p distances were calculated among haplotypes (Table 3). Haplotypes 1 and 2 differed by 1.2% (six substitutions, half of them being synonymous) and with the three haplotypes with p distances up to 10.4%. Distances between these three haplotypes ranged from 6.8% to 10.4% (from 33 to 52 substitutions). Lower p distances were obtained at the 774 bp alignment length with a minimum of 0.9% between haplotypes 1 and 2, and a maximum of 9.6% between haplotypes 2 and 4.

The five aforementioned haplotypes were distinguished based on phylogenetic analysis, genetic distances, and the ASAP species delimitation analysis. They were distributed within four clades, the first one including two separate sub-clades.

The first clade contained 83 sequences of *U. lacinulata* (including the lectotype specimen from Croatia, MW543061), 30 sequences labelled *U. 'laetevirens'*, 14 sequences labelled *U. 'rigida'*, 34 sequences identified as *Ulva* sp. A AF2021 according to *Fort et al. (2021b)*, one sequence from MNHN specimen of *U. 'rigida* var. *fimbriata'* (MT078957), one sequence from the MNHN specimen of *U. 'scandinavica'* (MT078956), the sequence from the holotype of *U. armoricana* (MT078955), and our haplotypes 1 and 2 (MT078946 and MT078950), supported with a 95% bootstrap value. Within this clade, p distances ranged

from 0 to 1.2% and the number of substitutions was less than 7. Haplotypes 1 and 2 were bound to separate subclades and were supported with 95 and 99% bootstrap values. The first subclade contained haplotype 1 (192 sequences), 83 sequences of *U. lacinulata* (including the lectotype) together with MNHN sequences of *U. armoricana*, *U. 'scandinavica'* and *U. 'rigida* var. *fimbriata'*, 30 sequences labelled *U. 'laetevirens'* and 14 sequences labelled *U. 'rigida'*, with a p distance ranging from 0 to 0.4% (0 to 2 substitutions). The second subclade presented a 0% p distance and clusters haplotype 2 (95 sequences) and 34 sequences identified as *U.* sp. A AF-2021 by *Fort et al. (2021b)* with MT160688, as a representative sequence. These two subclades were supported by ASAP as two separate species (500 bp alignment with 81 haplotypes, asap-score = 2.5 for 35 specific groups, *P*-val = 0.166; 774 bp alignment with 41 haplotypes, asap-score = 1.5 for 14 specific groups, *P*-val = 0.097).

The second clade contained one sequence of *U. 'rotundata'*, 21 sequences of *U. rigida* including the *U. rigida* lectotype specimen from Cadiz (MW543060) and nine sequences of our haplotype 3 (MT078951), supported with a 100% bootstrap value. Haplotype 3 presented 0% p distance with the *U. rigida* lectotype sequence (MW543060) produced by *Hughey et al. (2021b)*. This clade was supported as a single ASAP species partition using both the 500 and 774 bp alignments.

The third clade contained 181 sequences of *U. australis,* five sequences labelled as *U. 'pertusa'* and our haplotype 4 (MT078952, MT078953) with 41 sequences. The distance was from 0 to 5 substitutions (0.4% p distance) and supported with a 100% bootstrap value. This third clade was supported by ASAP as a single species by both the 500 and 774 bp alignments.

The last clade contained 180 sequences of *U. 'lactuca'* 32 sequences of *U. fenestrata* including the *U. fenestrata* holotype specimen (MK456404) and our haplotype 5 (MT078954). Sequences of this clade were distanced by three substitutions and a 0.6% p distance. This clade was supported by ASAP as a single species by both the 500 and 774 bp alignments.

## DISCUSSION

Our *Ulva*-specific *tuf*A primers allowed the amplification of this barcoding gene for 94% of sampled specimens but for only 27% of MNHN material. We identified five haplotypes attributed to nominal foliose *Ulva* species, and determined that haplotypes 1 and 2 are two distinct species. This determination is based on the General Lineage Concept of species (*De Queiroz, 1998*), on the analysis of intra- and interspecific genetic distances at *tuf*A (Tables 3 and 4), phylogenetic inference (Figure 2), and the ASAP species delimitation analysis.

On the ML tree (Fig. 2), haplotypes 1 and 2 clustered with sequences identified as *U. armoricana* (holotype), *U. lacinulata* (lectotype), *U. 'laetevirens'*, *U. 'rigida'*, *U. 'rigida* var *fimbriata'*, *U. 'scandinavica'* (closely matching haplotype 1) and *U.* sp. A AF-2021 (exact matches with haplotype 2) with p distances ranging from 0.4 to 1.4%. This range overlaps with interspecific p distances of 1.2 to 1.4% observed between both haplotypes and their nearest neighbour *U. 'taeniata'* (Table 4; 500 bp computations). This supports the view

**Table 4  Intra- and interspecific absolute (a) and uncorrected (p) distances for the 500 and 774 alignment lengths.** Closest sequences for the most closely related taxa on the *tuf*A ML tree with sequences representative of all known haplotypes. Data and closest sequence following *Kirkendale, Saunders & Winberg (2013)* (#). Dash ("-"): no p distance estimate due to limited sequence length for *U. taeniata* and *U. arasakii*.

| Haplotype, nominal species (number of sequences) | 500 bp this study | | | | | 774 bp this study and *Kirkendale, Saunders & Winberg (2013)* | |
|---|---|---|---|---|---|---|---|
| | Intraspecific | | Interspecific | | | | |
| | a | p (%) | Closest sequences | a | p (%) | a | p (%) |
| H1 *U. lacinulata* (Kützing) Wittrock (*n* = 192) | 0–2 | 0–0.4 | *U. taeniata* KC661445 | 6 | 1.2 | ≥ 7 | – |
| | | | *U. lactuca* lectotype of *U. lobata* MH730972, JN029303, # as *U. fasciata* JN029299 | 8 | 1.6 | 9 # | 1.16 # |
| | | | *U. sp. A AF-2021* MT160688 | 6 | 1.2 | 7 | 0.9 |
| H2 *U. sp. A AF-2021* (*n* = 95) | 0 | 0 | *U. taeniata* KC661445 | 7 | 1.4 | ≥ 10 | – |
| | | | *U. lactuca* MH763013 | 8 | 1.6 | 9 | 1.16 |
| | | | *U. lacinulata* MT160687, MT160689 | 5–7 | 1.0–1.4 | 7–9 | 0.9–1.16 |
| H3 *U. rigida* C. Agardh (*n* = 9) | 0 | 0 | *U. expansa* MH730973 Holotype | 22 | 4.4 | 33 | 4.3 |
| H4 *U. australis* (*n* = 41) | 0–2 | 0–0.4 | *U. arasakii* AB561079 | 26 | 5.2 | ≥ 35 | – |
| | | | *U. pseudorotundata* MT160686 | 34 | 6.8 | 50 | 6.5 |
| | | | *U. fenestrata* MK456404, MT160662, # as *U. lactuca* HQ610325, HQ610327 | 35–37 | 7.0–7.4 | 43 # | 5.56 # |
| H5 *U. fenestrata* (*n* = 2) | 0–3 | 0–0.6 | *U. arasakii* AB561079 | 17 | 3.4 | ≥ 19 | – |
| | | | *U. australis* MH538644, MT160679, MT160695 | 35–37 | 7.0–7.4 | 44-45 | 5.7–5.8 |
| | | | *U. rigida* MT160686 | 33 | 6.6 | 40 | 5.2 |

that p distances of 1.0 to 1.4% between haplotype 1 and haplotype 2 reflect interspecific diversity (Tables 3 and 4). *Kirkendale, Saunders & Winberg (2013)* calculated interspecific distances between 19 *Ulva* taxa, ranging from 0.65% (*U. ohnoi* JN029330 *versus U. lactuca* JN029303) to 5.56% (*U. australis* HQ610378 *versus U. fenestrata* HQ610325 as *U. 'lactuca'*) based on 774 bp sequences. If we focus on *U. lacinulata* (labelled *U. 'laetevirens'* in the study of *Kirkendale, Saunders & Winberg, 2013*), the minimum interspecific divergence is 1.16% with *U. lactuca* as delimited on 774 bp *tuf*A sequences (*Kirkendale, Saunders & Winberg, 2013*). Similarly, we found 1.16% divergence between *U. lactuca* (MH730972) and our haplotype 1 (Table 4; 774 bp computations). A range of 0.9 to 1.16% divergence was also estimated between *U. lacinulata* sequences previously labelled *U. 'laetevirens'* (MT160687 and MT160689) and our haplotype 2. This suggests that our delimited haplotypes 1 and 2, while very close genetically to *U. lacinulata*, represent two distinct species: *U. lacinulata* and an undescribed species *Ulva* sp. A, as proposed by *Fort et al. (2021b)*. This interpretation, based on phylogenetics and raw genetic distances, is supported by our results with ASAP, which evaluates intra- and interspecific genetic diversity within a coalescent framework. The complementary ASAP analysis performed on 136 *Ulva* sp. with complete *tuf*A sequences (1,224 bp) returned a partition (best asap-score = 4.0; *P*-val < 0.0002, 16 specific groups), which consistently clustered known conspecific taxa such as *U. australis/U. pertusa* (*Couceiro, Cremades & Barreiro, 2011*; *Hughey et al., 2021a*), *U. compressa/U. mutabilis* (*Steinhagen et al., 2019*), *U. lactuca/U. fasciata* (*Hughey et al., 2019*) as three distinct species. ASAP analysis also segregates *Ulva* sp. A from all other species. This ASAP analysis also supports the view that all *tuf*A sequences (1,224 bp long) previously labelled *U. 'laetevirens'* are molecularly identical or similar to *U. lacinulata* (MW543306), which is the lectotype specimen provided by *Hughey et al. (2021b)*. Consequently, haplotype 1 is considered to be *U. lacinulata* (Kützing) Wittrock, with *U. armoricana*, *U. rigida sensu* Bliding *non* C. Agardh and *U. laetevirens* sensu *Kraft, Kraft & Waller (2010)* as heterotypic synonyms. Haplotype 2 is considered to be an undescribed species, *Ulva* sp. A, following *Fort et al. (2021b)* for specimens collected along the European Atlantic coast (Ireland, UK and Portugal).

### Haplotype 1: *Ulva lacinulata* (Kützing) Wittrock 1882

The historical background of the description of *Ulva lacinulata* (Kützing) Wittrock was given by *Hughey et al. (2021b)*, revealing various and contradictory opinions on its taxonomy. Genetic analyses using ITS, *rbc*L and *tuf*A of the newly designated lectotype specimen (Herbarium Kützing L 0054997) however provided evidence that most, if not all sequences labelled as *U. 'armoricana'*, *U. 'scandinavica'* and *U. 'laetevirens'* were resolved in the same clade as the lectotype sequence of *U. lacinulata* (*Hughey et al., 2021b*).

*Ulva laetevirens* was first morphologically described in 1854 (*Areschoug, 1854*) with Port Philip Bay, Victoria, Australia as its type locality. *Areschoug (1854)* did not designate a holotype specimen and *Womersley (1984)* selected one of the two type specimens in Herb. Areschoug as a lectotype (S A2028 is a specimen with a large, expanded and lacerate frond). *Womersley (1984)* noted that lectotype cells "do not show the characteristics of *U. rigida*" but appear to be a "large, single frond with the cell dimensions and proportions

of *U. australis.*'' Accordingly, he placed *U. laetevirens* as a synonym of *U. australis*, which was not supported by subsequent studies (*Phillips, 1988*; *Kraft, Kraft & Waller, 2010*; *Kirkendale, Saunders & Winberg, 2013*), but is today validated by *Hughey et al. (2021a)* following molecular characterisation of the S A2028 lectotype of *U. laetevirens*. Through investigations of Southern Australian *Ulva* species, *Phillips (1988)* suggested that *U. rigida* C. Agardh and *U. rigida sensu* Bliding are two separate species hypotheses, the latter being referred as *U. laetevirens* when compared to Australian specimens (*Kraft, Kraft & Waller, 2010*). According to our results, all specimens from haplotype 1, together with a mixture of GenBank sequences labelled *U.* '*laetevirens*' and *U.* '*rigida*' (Fig. 2), did not match *U. australis* sequences (Table 3: 7–8% of inter-specific genetic divergence). Therefore, they are not conspecific with *U. australis* but are fully supported by the *U. lacinulata* lectotype (MW543061).

Our results highlighted low level of genetic variability amongst *tuf*A sequences of the *U. lacinulata* group (Fig. 2). This variability was already noticeable from the results of *Kirkendale, Saunders & Winberg (2013*, *tuf*A sequences as *U.* '*laetevirens*'), *Miladi et al. (2018)*, *tuf*A sequences as *U.* '*laetevirens*'), *Steinhagen, Karez & Weinberger (2019*, *tuf*A sequences as *U.* '*rigida*') and *Fort et al. (2021a*, sequences labelled as *U.* '*laetevirens*'). The representative sequences of the first subgroup were *U.* '*laetevirens*' HQ610428 (sampled from BC, Canada, but initially labelled *U.* '*rigida*' by (*Saunders & Kucera, 2010*) and JN029322 (sampled from North Brighton in the vicinity of the type locality of *U.* '*laetevirens*' at Port Phillip within Port Phillip Bay, Victoria, Australia). This subgroup included sequences labelled *U.* '*laetevirens*' sampled from Connecticut, USA (*Mao et al., 2014*), the Wadden Sea in Germany (*Steinhagen, Karez & Weinberger, 2019* as seven sequences labelled *U.* '*rigida*'), Italy (*Wolf et al., 2012* as *U.* '*rigida*' HE600178 to HE600182; *Miladi et al., 2018*), and Tunisia (*Miladi et al., 2018*). Representative sequences of the second subgroup were JN029321, JN029324, JN029325 and JN029327 from Australia (*Kirkendale, Saunders & Winberg, 2013*). Our 192 analysed sequences of haplotype 1, together with the 3 MNHN sequences of *U. armoricana* (holotype), *U.* '*rigida* var *fimbriata*' and *U.* '*scandinavica*', were also included in this subgroup (Fig. 2).

The *Ulva armoricana* holotype specimen MNHN-PC-PC0115137 was collected at Roscoff in 1996 and analysed using ITS with a sequence referred to as 'U. arm.8' (MT078962) by *Coat et al. (1998)*. A Blastn analysis (*Zhang et al., 2000*) revealed that 'U. arm.8' together with the sequence labelled as 'U. arm.2' (MT078963) presented a 99.44% similarity (3 substitutions of difference) with the ITS sequence of *U. lacinulata* (MW544060) provided by *Hughey et al. (2021b)*. Similarly, the *rbc*L sequence of *U. armoricana* holotype (MT078960, Supplemental S1) presented a 99.99% similarity (one substitution of difference) with the *rbc* L sequence of *U. lacinulata* (MW543061). This confirms that the holotype specimen of *U. armoricana* is identical or nearly identical to *U. lacinulata* using the legacy markers ITS, *rbc*L and *tuf*A.

The *Ulva* '*scandinavica*' specimen MNHN-PC-PC0547277 was collected in Brittany by R. Kuhlenkamp and determined following *Hoeksema & Van den Hoek (1983)*. The description of *U.* '*scandinavica*' by these authors does not match the concept of *U. scandinavica* given by *Bliding (1969)* and should be regarded as *U. rigida* sensu Bliding (B. de Reviers, personal

communication). Our results support this opinion. Numerous studies have suggested conspecificity between specimens determined as *U. 'scandinavica'* and *U. 'laetevirens'* or *U. 'rigida'*, on the basis of ITS (*Shimada et al., 2003*; *Hayden & Waaland, 2004*; *Mao et al., 2014*), *rbc*L (*Loughnane et al., 2008*; *Wolf et al., 2012*; *Wan et al., 2017*; *Hughey et al., 2021a*), or both ITS and *rbc*L (*Hayden & Waaland, 2004*; *Kraft, Kraft & Waller, 2010*). Recent molecular analyses of *U. rigida* and *U. lacinulata* lectotype specimens together with *U. scandinavica* material from Bliding have clarified synonymies of these species (*Hughey et al., 2021b*) and support the conspecificity between *U. scandinavica* and *U. lacinulata* (*Fort et al., 2021b*; *Hughey et al., 2021b*).

Ulva rigida var. fimbriata J. (*Agardh, 1883*) is regarded by *Phillips (1988)*, p. 440-443) as a synonym of *U. 'laetevirens'* based on the examination of cell conformation in type and holotype specimens. Our results agree with this assessment, indicating that the specimen *U. rigida* var. *fimbriata* MNHN-PC-PC0531492 collected in La Coruna, Spain, belongs to the clade supported by the lectotype specimen of *U. lacinulata* (Fig. 2). This clade includes several sequences labelled *U. 'laetevirens'* (*Saunders & Kucera, 2010*; *Fort et al., 2021a*) and/or identified as *U. lacinulata* (*Fort et al., 2021b*). *Ulva rigida* var. *fimbriata* is only reported from the Atlantic coasts of Spain and Portugal (*Gallardo et al., 1993*; *Guiry & Guiry, 2021*). Transverse sections of the basal regions of the thallus of specimens collected from western Portugal (*Lima et al., 2017*) conform with *U. 'laetevirens'* cell shape descriptions given by *Kraft, Kraft & Waller (2010)*, *Sfriso (2010)* and *Mao et al. (2014)*, with an elongated, narrow, conical shape which is the opposite of the large and rectangular shapes observed in basal regions of *U. rigida* C. Agardh. A molecular characterization of the type specimens located at Lund Herbarium (LD14324 and LD14325) is needed to confirm the heterotypic synonymy with *U. lacinulata*.

## Haplotype 2: *Ulva* sp. A *Fort et al. (2021b)*

*Ulva* sp. A is an undescribed species previously labelled as *U. 'rigida'* by *Fort et al. (2021a)* but separated from *U. lacinulata* by a general mixed yule coalescent model (GMYC) analysis using *tuf*A and ITS1 together with comparison of their respective organellar genomes (*Fort et al., 2021a*; *Fort et al., 2021b*). Our ASAP analyses using partial (500 and 774 bp) and complete 1,224 bp *tuf*A sequences support this view (Table 4 and Supplemental S9). However, no *U. lacinulata*/*U.* sp. A separation was noted using *rbc*L (*Fort et al., 2021b*). This low genetic variability leads *Hughey et al. (2021b)* to suggest conspecificity between *U. lacinulata* and all related *U. 'rigida'*. Both *Hughey et al. (2021b)* and *Fort et al. (2021a)*, *Fort et al., 2021b*) have used the *rbc*L sequence AY422564 from a Chilean *U. 'rigida'* specimen (voucher WTU344827 from Pelluco Beach) in their respective analyses of the *U. lacinulata*/*U. 'rigida'* group. This *rbc*L sequence was used by *Fort et al. (2021a)* as a reference sequence to specifically support all the sequences labelled *U.* sp. A (*Fort et al., 2021b*). It is noticeable that the ITS sequence (AY422522) of this voucher, as analysed by *Hayden & Waaland (2004)*, is 100% identical to the ITS rDNA sequence (AY260565: ITS1-5.8S-ITS2 of 515 bp) of another specimen labelled *U. 'rigida'* (voucher WTU 344826 from the Burke Museum), which was collected from Cadiz (*Hayden et al., 2003*; *Hayden & Waaland, 2004*). Additionally, all the three *U. 'rigida'* sequences (*U. rig.* 1–3) from

specimens collected in Brittany by *Coat et al. (1998)* have ITS sequences (MT078965, MT078966 and MT078967) identical in ITS to the Spanish *U. 'rigida'* (AY260565). These comparisons using ITS may suggest that the geographic distribution of *U.* sp. A. is larger than previously estimated by *Fort et al. (2021b)*, and include not only Ireland, UK and Portugal but also the Atlantic coast of France and Spain together with the Pacific coast of South America.

### Haplotype 3: *Ulva rigida* C. Agardh 1823

*Ulva rigida* was described in 1822 by C. *Agardh (1823)* with a geographic distribution from the Atlantic Ocean (including the Cape of Good Hope) to the Mediterranean and Black Seas. Agardh's son, J.G. Agardh, provided detailed coloured drawings of the cellular morphology of *U. rigida* (*Agardh, 1883*), see his Table IV and figure 19–122). Although C. Agardh did not assign a holotype specimen to the type series placed in LD (Lund Herbarium, Sweden), a lectotype (LD14294) was designated by Papenfuss in 1940 (*Papenfuss, 1960*), see p. 305 his Plate 1 and figure 11). His choice was based on one of the two specimens collected by Cabrera on the Atlantic coast of southern Spain. *Papenfuss (1960)* deduced from Cabrera's practices in phycology that the lectotype came from Cadiz, Spain. According to *Ricker (1987)*, another specimen (LD14449) was independently selected by R.B. Searles in 1975 for lectotypification, but this remained unpublished (*Guiry & Guiry, 2021*). The lectotype specimen (LD14294) was molecularly characterised for ITS, *rbc*L and *tuf*A by *Hughey et al. (2021b)*, suggesting that all sequences labelled *U. 'pseudorotundata'* in Europe are mislabelled and are identical or nearly identical to *U. rigida*.

Haplotype 3, with nine samples from Concarneau, presents 22 substitutions (4.4% p distance) with the closest clade, composed of two sequences of *U. expansa* (Setchell) Setchell & N.L. Gardner including the holotype specimen (MH730973), and 30 sequences labelled *U. 'lobata'* (Kützing) Harvey. According to *Hughey et al. (2019)*, these *U. 'lobata'* sequences sampled in the Northeast Pacific, should be named *U. expansa* because of the synonymy of the *U. expansa* holotype and *U. 'lobata'* sequences from the northeast Pacific, based on *tuf*A and *rbc*L analyses. The maximum intraspecific p distance is 0% for the *tuf*A gene among the 32 GenBank samples of *U. expansa*. The 4.4% p distance between these sequences and our haplotype is also too large to consider our haplotype to be within the intraspecific range of *U. expansa*. Haplotype 3 clustered with the sequence of the *U. rigida* lectotype together with 21 sequences of *U. rigida* previously labelled *U. 'pseudorotundata'* collected in Ireland and Portugal by *Fort et al. (2019)* and *Fort et al. (2021a)*. To strengthen our taxonomic interpretation of this haplotype, we sequenced a short part of the *rbc*L gene typically used in museum type analyses (*Hanyuda & Kawai, 2018*). The Blastn analysis (*Zhang et al., 2000*) of two samples of our haplotype 3 (MW013545, 238 bp long) revealed a 99.58% similarity (one substitution of difference) with sequences labelled *U. 'pseudorotundata'*, *U. 'rotundata'* and the lectotype of *U. rigida*. *Ulva 'pseudorotundata'* has been reported in Roscoff as *U. 'rotundata'* (*Hoeksema & Van den Hoek, 1983*). The synopsis of *Hoeksema & Van den Hoek (1983)* was used by *Coat et al. (1998)* in describing ITS sequences of specimens collected at Roscoff in 1994–1995 and morphologically attributed to *U. 'rotundata'*. It was further demonstrated (*vide infra*) that these sequences

were attributable to *U. australis* (*Couceiro, Cremades & Barreiro, 2011*). Consequently, the presence of *U. rigida* at Roscoff (see *Dizerbo & Herpe, 2007*; *Loiseaux-de Goër & Noailles, 2008*) cannot be confirmed by our results (Table 3). However, the current reports of *U. rigida* in Concarneau may add a new record of the species for southern Brittany (*Dizerbo & Herpe, 2007*; *Burel, Le Duff & Gall, 2019*). The species has also been described in Ireland in green tide (*Wan et al., 2017*; *Fort, Guiry & Sulpice, 2018*) and non-green tide contexts (*Fort et al., 2020*) as *U. 'rotundata'* and/or *U. 'pseudorotundata'*.

## Haplotype 4: *Ulva australis Areschoug, 1854*

Haplotype 4 was reported from two sites along the Brittany coasts (Concarneau and Roscoff) and clustered with many sequences of *U. australis* and *U. pertusa* on *tuf*A gene analysis, with a p distance below 0.4% on 500 bp. A similar result was obtained by *Lee, Kang & Kim (2019)*, who determined the intraspecific variation at *tuf*A (ca 800 bp) in the range 0–0.4% for *U. australis* from Jeju Island, Korea, within the native distribution area of the species. *Kirkendale, Saunders & Winberg (2013)* determined a minimum interspecific divergence of 5.56% with *U. fenestrata* (as *U. 'lactuca'*) based on 774 bp, compared to 6.8% on our 500 bp for *U. australis*. Based on these values, haplotype 4 presents a p distance within the intraspecific range of *U. australis*. A complementary comparison of full-length *tuf*A gene sequences (1,224 bp) of *U. australis* and *U. fenestrata* (*Hughey et al., 2019*; *Fort et al., 2021a*) revealed that their interspecific genetic distance was limited to 4.6 to 4.7%, based on 56 to 58 SNPs.

*Ulva australis* was described in 1851 at Port Adelaide, South Australia (*Areschoug, 1854*). *Phillips (1988)* included *U. australis* within the *U. rigida* C. Agardh taxon based on morphological and developmental characteristics. However, *Kraft, Kraft & Waller (2010)* excluded it from this taxon and considered *U. australis* as a species of its own. *Kjellman (1897)* described *U. pertusa* from three localities in Japan independent of observations by *Areschoug (1854)*. A more recent comparative study based on the analysis of *rbc*L and ITS1 sequences suggested that *U. australis* from Southern Australia and *U. pertusa* from Japan are conspecific and widely distributed, as an introduced species, along Iberian coasts (*Couceiro, Cremades & Barreiro, 2011*). *Ulva pertusa* Kjellman is recognised today as a heterotypic synonym of *U. australis* (*Guiry & Guiry, 2021*). Molecular analysis of the lectotype of *U. australis* (*Hanyuda & Kawai, 2018*) together with one lectotype and two syntypes of *U. pertusa* (*Hughey et al., 2021a*) supported this synonymy. *Hanyuda & Kawai (2018)* further suggested that populations of *U. australis* are non-indigenous in Australia but were introduced from northeast Asia and not directly from Japan by the middle of 19th century. *Ulva australis*, as *U. pertusa*, has been reported throughout the world, including the Mediterranean Sea since the early 1970s (*Verlaque, Belsher & Deslous-Paoli, 2002*; *Hanyuda et al., 2016*). This species had been reported in Brittany, at Roscoff, from October 1994 to October 1995 by *Coat et al. (1998)* as misidentified specimens of *U. 'rotundata'* (*Couceiro, Cremades & Barreiro, 2011*) and at Beg Meil, near Concarneau, in 2018 by *Fort et al. (2020)* and *Fort et al. (2021a)*. These authors also reported the species from several Brittany localities (Lannion Bay and Brest), suggesting that *U. australis* may be a common inhabitant of West Brittany coasts and a major contributor to local green tides

(*Fort et al., 2020*). The last synopsis of French records, on the basis of morphology records (*Verlaque, Belsher & Deslous-Paoli, 2002*) and molecular data, suggests that this species is largely overlooked along the French Atlantic coasts (*Sauriau et al., 2021*). At the end of the 20th century, the port of Concarneau was the third biggest tuna fishery port in France (*Couliou & Piriou, 1989*) with many ships involved in worldwide tuna fisheries. This makes marine algae communities in the vicinity of Concarneau particularly vulnerable to the introduction of non-native species such as *U. australis.*

## Haplotype 5: *Ulva fenestrata* Postels & Ruprecht 1840

Haplotype 5 was detected only twice from Concarneau. It clustered with many sequences of *U. 'lactuca'* and *U. fenestrata*, including a sequence from the holotype of *U. fenestrata* MK456404 (*Hughey et al., 2019*). Uncorrected-p distances range from 0 to 0.6% with three substitutions. We hypothesize that haplotype 5 belongs to the *U. fenestrata* group considering its p distance of 3% with all *U. 'arasakii'* sequences (all identical to AB561079, Fig. 2).

Ulva lactuca has been described by *Linnaeus (1753)* who did not designate a type specimen. The specimen marked '5' in the Linnaean herbarium has been recognised as the type *U. lactuca* by *Papenfuss (1960)*, based on the analysis of the Species Plantarum (*Linnaeus, 1753*). However, further examination revealed a difference with the modern taxonomic hypothesis for *U. lactuca*. This specimen had marginal teeth on the thallus margin, unlike the description of the current *U. lactuca* from Europe. Following *Papenfuss (1960)*, *Bliding (1969)* also identified this type as a sample that may have been collected on the Swedish west coast. This hypothesis was later rejected by *Hughey et al. (2019)*. The *U. lactuca* holotype was molecularly analysed by *Hughey et al. (2019)* revealing that the *U. lactuca* described by Linnaeus is called *U. fasciata* Delile in the subtropics, and *U. lobata* in the eastern Pacific Ocean. The lectotype of *U. lobata* (Kützing) Harvey was renamed *U. lactuca* (*Hughey et al., 2019*). These authors also found that European *U. 'lactuca'* *rbc*L sequences clustered with the *U. fenestrata* Postels & Ruprecht holotype sampled in eastern Russia, in Avacha Bay. This suggests that all of the 180 *U. 'lactuca'* *tuf*A sequences within the group of the *Ulva fenestrata* holotype (MK456404) should be *U. fenestrata*. Many authors have already suggested conspecificity of *U. 'lactuca'* and *U. fenestrata* (*Hayden et al., 2003*; *Hayden & Waaland, 2004*; *Loughnane et al., 2008*). *Ulva fenestrata* was reported in locations that include the Pacific Ocean, in Washington state (*Nelson, Nelson & Tjoelker, 2003*), and in Europe (*Hughey et al., 2019*). It has been reported in Beg Meil near Concarneau as *U. 'lactuca'* (*Fort et al., 2020*) and later as *U. fenestrata* (*Fort et al., 2021a*). Indeed, the *rbc*L sequence AB097622 of *U. 'lactuca'* used by these authors was identified as *U. fenestrata* by *Hughey et al. (2019)*.

## Potential issues with type specimens

Careful consideration must be given to GenBank sequences which species names were assigned based on morphology. This was previously demonstrated for many *Ulva* species such as *U. fasciata, U. fenestrata, U. lactuca, U. laetevirens, U. lobata, U. pertusa, U. spathulata U. stipitata*, and *U. tenera* (*Hanyuda & Kawai, 2018*;

*Steinhagen, Karez & Weinberger, 2019*; *Hughey et al., 2019*; *Hughey et al., 2021a*). Assuming that the *tuf*A gene trees represent species trees within the genus, our study revealed some potential issues with the identification of *Ulva* sequences on GenBank. For instance, *U. 'laetevirens'* LT969813 and *U. 'rigida'* KC661447 do not match any of our haplotypes (Fig. 2) and could not be attributed to a validly named species with the support of museum type materials. Similarly, *U. fasciata* sequences are considered to be a synonym of *U. lactuca* (*Hughey et al., 2019*) that formed a paraphyletic assemblage along with other taxa. Two sequences of *U. 'reticulata'* identical to MG963806, together with the sequence KC661468, and each *U. 'reticulata'* and *U. 'fasciata'* sequence of this clade should be renamed *U. lactuca* based on the 0% p distances between *U. lactuca* MH730972 and the 24 *U. 'fasciata'* sequences identical to JN029299. All other sequences labelled *U. 'fasciata'* used in our analysis (Fig. 2) are thus misidentified. *Fort et al. (2021b)* provided evidence that misidentification of GenBank sequences is not restricted to a few *Ulva* species but is inherent to the taxonomic studies of the *Ulva* genus. Finally, the inclusion of museum types in taxonomic analysis, as previously stated by *Loughnane et al. (2008)*, allows major clarification of the taxonomy of the *Ulva* genus (*Hughey et al., 2019*; *Hughey et al., 2021a*; *Hughey et al., 2021b*). From this point of view, analysis of the holotype materials of *U. gigantea* (Kützing) Bliging 1969 (type material located at Lund Herbarium) should strengthen results of further studies of foliose *Ulva* taxa.

In addition to these taxonomic issues, and as suggested above, lack of resolution of chloroplastic and nuclear-ribosomal molecular markers may cause confusion. What has been identified as intra- and interspecific variation at *tuf*A may not reflect true evolutionary history. There is a strong need to integrate data from the morphology, physiology, ecology, and different types of molecular markers in order to delineate species for this and other taxonomic groups. In *Ulva*, the sequencing of restriction-site associated DNA (RAD-seq) has proven feasible and produced data that are partially incongruent with *rbcL* barcoding (*Fort, Guiry & Sulpice, 2018*). Similarly, *Fort et al. (2021a)* and *Fort et al. (2021b)* promoted the use of the complete cytoplasmic genome (mitochondrion and chloroplast) to compare species and estimate intra- and interspecific genetic divergence. Other types of molecular markers, such as *trn*A-N or *atp*I-H regions, could provide information on the spatial patterns of genetic diversity and biogeography, as exemplified by *U. australis* on a worldwide scale (*Hanyuda et al., 2016*) and along the French coasts (*Sauriau et al., 2021*). These markers may aid in testing the autochthonous/allochthonous status of other *Ulva* species, particularly for specimens labelled *U. 'laetevirens' sensu* Kraft, Kraft & Waller, which may be introduced from Australasia (*Kirkendale, Saunders & Winberg, 2013*; *Mao et al., 2014*). The current synonymy with *U. lacinulata*, as evidenced by *Hughey et al. (2021b)*, opens new testable hypotheses since the species was primarily described from the Adriatic Sea (*Kützing, 1847*).

## CONCLUSIONS

This study confirms the presence of five foliose *Ulva* species that had been misidentified using morphology alone along Brittany and Vendée coasts. These findings are in agreement

with those in *Fort et al. (2020)*, *Fort et al. (2021a)*and *Fort et al. (2021b)*, and add some molecular supports for the taxonomic review by *Burel, Le Duff & Gall (2019)*. The current report of *U. australis*, which was introduced from north eastern Asia (*Hanyuda et al., 2016*; *Sauriau et al., 2021*), is congruent with earlier results by *Coat et al. (1998)* at Roscoff, Brittany. Identification of *U. armoricana* was challenged by sequencing *tuf*A and *rbc*L markers for the holotype specimen from Roscoff. As a consequence, the status of *U. armoricana* as a heterotypic synonym of the oldest valid name *U. lacinulata* (Kützing) Wittrock is confirmed. Additional sampling during bloom seasons (summer and early fall) will advance the study of the specific composition of green tides along the French coasts, and the respective roles of these *Ulva* species in such phenomenon. New investigations using molecular analyses of museum type materials may shed light on these issues.

## ACKNOWLEDGEMENTS

We are grateful to Dr. Bruno de Reviers, Muséum national d'Histoire naturelle (MNHN) Paris, for his advice and comments, and to Nadia Améziane, director of the Marine Station at Concarneau (MNHN) for providing access to the Marine Station laboratory facilities. We also thank Aya Ghedmasi and Salomé Ducos for help with laboratory work, and Fabien Aubert, Valérie Huet and Michel Prineau for help with sampling and laboratory work. We thank Stacy Krueger-Hadfield for her advice on the study. The authors thank the molecular core facility of the LIENSs laboratory and Lionel Kervran of the cryptogam collection of the MNHN, for sampling the herbarium. The authors gratefully acknowledge Marc Costello, Antoine Fort, Giovanni Furnari, Ronan Sulpice, Florian Weinberger, two anonymous reviewers, and the PeerJ copyediting service, whose comments greatly improved our manuscript.

### Funding

This research was supported by the CNRS, La Rochelle Université, Parc naturel régional du Marais poitevin (contract 180166-04/04/2018), the project DEVOTES (DEVelopment Of innovative Tools for understanding marine biodiversity and assessing good Environmental Status, FP7, grant no. 308392), and the project ECONAT funded by the Contrat de Plan Etat-Région, CNRS and the European Regional Development Fund (FEDER). The funders had no role in study design, data collection and analysis, decision to publish, or preparation of the manuscript.

### Grant Disclosures

The following grant information was disclosed by the authors:
The CNRS, La Rochelle Université, Parc naturel régional du Marais poitevin (contract 180166-04/04/2018).
The project DEVOTES (DEVelopment Of innovative Tools for understanding marine biodiversity and assessing good Environmental Status, FP7): 308392.

The project ECONAT funded by the Contrat de Plan Etat-Région, CNRS and the European Regional Development Fund (FEDER).

## Competing Interests

The authors declare there are no competing interests.

## Author Contributions

- Manon Dartois and Eric Pante conceived and designed the experiments, performed the experiments, analyzed the data, prepared figures and/or tables, authored or reviewed drafts of the paper, and approved the final draft.
- Amélia Viricel analyzed the data, authored or reviewed drafts of the paper, and approved the final draft.
- Vanessa Becquet performed the experiments, authored or reviewed drafts of the paper, and approved the final draft.
- Pierre-Guy Sauriau conceived and designed the experiments, analyzed the data, authored or reviewed drafts of the paper, contributed funding, and approved the final draft.

## Field Study Permissions

The following information was supplied relating to field study approvals (i.e., approving body and any reference numbers):

Specimens of green algae were sampled at three sites along the French Atlantic coast. Sampling of this species is not regulated and does not require permits.

## DNA Deposition

The following information was supplied regarding the deposition of DNA sequences:

*tuf*A haplotypes MT078946, MT078947, MT078948, MT078949, MT078950, MT078951, MT078952, MT078953, MT078954, MT078955, MT078956 and MT078957, and rbcL haplotypes MT078960 and MT078961 produced in this study are available at GenBank.

## Data Availability

The code for generating the ML tree and R code for preparing the sampling map are available Supplemental Files.

## Supplemental Information

Supplemental information for this article can be found online at http://dx.doi.org/10.7717/peerj.11966#supplemental-information.

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
