# Peer review of "Molecular genetic diversity of seaweeds morphologically related to Ulva rigida at three sites along the French Atlantic coast"

_PeerJ, doi:10.7717/peerj.11966_

## Round 0.1 · original submission · Major Revisions

I am pleased to have received 3 constructive reviews. Please address them and improve the paper. I note some use the phrase green tides which seems inappropriate because Ulva are benthic, not floating tides. The term bloom is sometimes used but indicates problematic growth which may not necessarily be the case in all these situations. A lot of Ulva may be due to lack of herbivores, or salinity effects on grazers for example.

·

Basic reporting

As correctly indicated by the tile this paper describes the molecular genetic diversity of seaweeds morphologically related with Ulva rigida at three sites along the French Atlantic coast.
The language needs improvement and I provide some (incomplete) suggestions in a file attachment.
The field background is correctly provided. A very recent relevant reference might be included (see below).
The article structure is correct and figures and tables as well. Raw data are shared, including even the R scripts that were used.
The results shown are relevant to support the conclusions made.

Experimental design

This paper presents and discusses original primary research data that are within the scope of PeerJ.
The introduction clearly describes knowledge gaps that are adressed and filled by this paper.
The methodology and the statistical treatment are rigorous and up to date, and described in detail to allow for replication by other researchers.

Validity of the findings

I have no doubt about the validity of the findings. Relatively large numbers of replicates have been analyzed at each of the sites that were investigated.
The underlying data are provided and their treatment is described with more detail than in many other similar studies.
The conclusions are clearly supported by the results and the authors take care to prevent overinterpretations.
.

Additional comments

Dear authors,
although this is not a criterion for PeerJ I wish to tell you that I found your study interesting...I feel the language needs some improvement and I provide a file attachment with some suggestions. However, please be aware that I am also not a native English speaker and thus not the best suited person to help in this respect.
A very recent publication that I think is relevant in the context of your work is Fort et al. (2020) in J Phycol (see https://doi.org/10.1111/jpy.13079), who also studied (among others) samples from Brittany and found (limited) evidence for hybridization between U. rigida and U. laetevirens.
Florian Weinberger

Reviewer 2 ·

Basic reporting

This study examines the genetic diversity of green tide-forming Ulva along the Atlantic coast of France based on tufA sequence data from a large number of samples. The main conclusions are that these green tides are formed by four different taxa, including U. rigida, the introduced U. australis, U. fenestrate, and (possibly) U. pseudorotundata. In addition, some taxonomic issues are tackled, such as the synonymy of U. armoricana with U. rigida. The manuscript is generally well presented, although there is room for improvement.

The aim and main focus of the study should be stated more clearly. It is unclear if the study focusses on Ulva diversity in general, on diversity of green tides, or on the diversity within the morphotype U. rigida? If U. rigida is the focus of the paper, then the context of this taxon should be better explained in the Introduction. Now, U. rigida is only mentioned marginally in the Introduction. Along the same line, the Material and methods section should state explicitly how sampling was performed: were U. rigida samples only identified in the field, or were identifications also based on microscopy in the lab? Were only free floating (green tide-forming) individuals sampled, or also individuals that were attached to the substrate?

The morphology and taxonomy of genus Ulva has been extensively studied in Europe in the 1960-ies to 1980’ies (papers by Bliding, Koeman, …). This should be mentioned in the Introduction. Now, only the statement “Taxonomic studies of this taxon are mainly based on molecular phylogenies” is made, which is clearly incomplete.

Experimental design

The authors should state more clearly how samples were identified (only in the field, or also based on microscopy) and sampled (only free floating/green tide forming Ulva individuals, or also attached individuals). Additional sampling parameters should be provided, such as intertidal/subtidal, coastal distance that have been sampled, etc.

Only a single marker is used in this study, tufA. In the Introduction, the authors argue that “most Ulva species have been identified on tufA”. However, many recent studies have used data from additional markers, including rbcL and ITS rDNA. In addition, sequencing studies of historic type material have mainly relied on the rbcL gene (Hughey et al 2020). It is therefore unfortunate that this study only includes tufA data. I believe a better argumentation should be given on why only tufA was sequenced, or preferably that additional markers are sequenced, at least for a selection of samples based on the different tufA haplotypes.

It is unclear how species clusters were delimited. In the Discussion it is briefly mentioned that some kind of threshold between intra- and inter-specific genetic distances was used, and the General Lineage Concept of species is mentioned, but details of these analyses are lacking. This should be clearly explained in the Methods and Results sections.

Validity of the findings

Details on species delimitation methods should be provided and reported in the Results section.

Additional comments

See 1. Basic reporting.
In addition, there are numerous language issues, so the entire manuscript should be carefully checked for spelling and grammatical errors.

Reviewer 3 ·

Basic reporting

This study aims to provide a detailed picture on the molecular diversity of leaf-like Ulva species associated with green tides along the French Atlantic coast. One of the main findings of this study is that five tufA haplotypes which were allocated to four different species are present at green tide hot-spots.
Furthermore, the authors including historic herbarium specimens to underpin the synonymity of Ulva armoricana and Ulva rigida. The manuscript is generally well written and illustrated. However, the main figure (figure 2) would benefit from larger fonts for the bootstrap values and from highlighting the respective discussed clades (haplotypes 1-5).

Experimental design

Ulva species are common and widely spread around the globe. In most of the cases they find their vegetation peak during the summer months. It is therefore unfortunate that only data from early spring (January/ February) is provided. However, the restricted sampling remains unexplained by the authors. In general the study is based on a solid sampling size (360 specimens) but excluding seasonality could be problematic due to green tides having their main abundance during summer (June-August).

One of my main concerns is that the methodologies for species delimitation are not explained in detail. The authors state that species delimitation is based on molecular data, but do not include distinct delimitation methods such as GMYC models to break down difficult relationships like within the Ulva rigida-laetevirens cluster.

Another concern is that only one marker (tufA) has been used. However, as correctly explained by the authors in the introduction, data sets are in most of the cases either built on rbcL or tufA which makes comparisons difficult. Furthermore, the tufA gene is around 770+ bp long but the data set used in this study has been trimmed to 500 bp. The cut-off of 1/3 of the sequences is not explained by the authors.

Validity of the findings

As stated above, the study would have largely benefited from including a seasonal sampling in summer when green tides have their peaks. Additionally, the proposed synonymity of U. rigida and U. armoricana should be done in a formal way (naming homotypic and heterotypic synonyms, including morphological characters etc.). However, due to uncertainties of the taxonomic status of U. rigida and U. laetevirens I find it hard to base a synonymisation on tufA sequences only and would rather suggest to combine several markers or genome sequences/ cytoplasmic genome. The status of U. rigida/ U. laetevirens has been intensely discussed by Fort et al 2020 (https://onlinelibrary.wiley.com/doi/abs/10.1111/jpy.13079).

Additional comments

Introduction

Line 49: ” These species are considered as pioneer and opportunistic organisms”. Perhaps better: “some of their representatives….” since several Ulva species do not tend to proliferate.

Line 50: The genus in general has a global distribution, not all Ulva species.

Line 58: Be more specific here. The tragic deaths were not caused by the algal mats degradation itself but by its toxic gases that were released.

Line 80: I disagree. The largest database for genetic identification within the genus Ulva is still based on the rbcL gene. However, as explained below, tufA offers a more “reliable” marker.

Line 82: I agree. However, you are using a smaller fragment (~500bp) of the tufA gene in this study even though the gene itself contains 770+ bp. Why is the used fragment trimmed so much? You are losing nearly a third of information here.

Line 86: “This paper aims…” perhaps better “Our/This study aims…..”


Material and Methods:

Line 99: The sampling is restricted to spring only. Since it is known that the main green tides occur during the early to late summer months (June-August) it is doubtful that the sampling reflects the summer conditions (e.g. species appearance, amount, abundance). That the specimens were in early life-stages gets also reflected by their rather small size (line 104, 7cm).

Line 119: Can you explain why you designed new primers? Please state the length of the amplicon in comparison to Saunders and Kucera 2010. Furthermore, the table says the amplicon length is 800 bp why is the analysis performed on trimmed sequences of 500 bp lenth only?

Line 128: As stated in the introduction it is a pity that either tufA or rbcL sequences are available per data set. This study would have largely benefited from including rbcL sequences as well. By including another marker like rbcL this study would have had a more global impact and could also be connected to past results based on rbcL only.

Line 136: Please state the length of your amplicon in comparison to the significant longer reference sequences.

Line 144: How was the evolutionary model determined?

Line 148: Typo - Ulva compressa

Line 148: U. compressa as well as U. intestinalis have been found as leaf-like as well and are not obligatory tubular. Furthermore, both species have been found as essential parts of green tides.

Line 152. Why was no GMYC method applied. You are presenting the haplotypes here but it would be nice to have an estimation of the haplotypes per species which could be perfectly matched with a solid GMYC analysis. This would also give more insights into the species delimitation of the U. rigida/ U. laetevirens case.


Results:

In general I find it hard to follow the results. It would be easier if you would mark the respective clades which represent the haplotypes discussed in the text with a dashed box in figure 2. Although you are giving the numbers of sequences of one clade in brackets, it is not really clear if those sequences are the ones listed in table 1 and 2? Additionally, why is only one phylogenetic method used in the phylogenetic tree? I suggest to include at least the Bayesian method as well. The bootstrap values are hard to read.
Additionally, it would be great if morphological data on the sampled individuals would be made available. Is the longer alignment showing differences in the phylogenetic analysis compared to the trimmed (500bp) alignment?

Line 179: Typo – File



Discussion:

Line 205: The history of U. rigida/ U. laetevirens is nicely presented. However, Fort et al. 2020 (https://onlinelibrary.wiley.com/doi/abs/10.1111/jpy.13079) should be cited and discussed in detail. Your and above mentioned study are in a way contradictory when it comes to the species concept of U. rigida/ U. laetevirens. This could be problematic when suggesting the synonymy of U. rigida and U. armoricana.

---

## Round 0.2 · Major Revisions

The referees note the improved revisions but also that they have not checked the language sufficiently and an important point about the title and key message. Indeed, only 3 sites at the same time of year may not provide sufficient sampling to determine the "Molecular genetic diversity of the macro-morphological taxon Ulva cf. rigida along French Atlantic coasts". This genus is famous for its variety of growth forms. Might hybrids of locally selected genotypes be possible? Please give this a much more rigorous self-critique of what the data can and may tell us, and of course a better quality of writing.

Reviewer 2 ·

Basic reporting

The authors have adequately revised their manuscript according to my previous comments. I don't have any further fundamental concerns. The text itself is still rather sloppy with numerous errors. Below is a non-exhaustive list. The authors are encouraged to carefully go through the entire text to correct spelling and grammatical errors.

Line 34 ‘genetic analysis at tufA’ -> of tufA
Line 108 ‘This intriguingly questions’ -> These intriguing questions
Line 112 ‘The conspecifity with U. rigida’ -> conspecificity
Line 185 ‘individual numbered plastic bag’ -> individually numbered plastic bags
Line 348 ‘and could not be further used to described’ -> to describe
Line 351 ‘However, these sequences should be temporary considered’ -> temporarily
Line 359 ‘As Areschoug did not designated’ -> designate
Line 469 ‘sequences of specimen collected at Roscoff’ -> specimens
Line 508 ‘on the basic of morphology’ -> on the basis of
Line 543 ‘U. stipiata’ -> stipitate?
Line 583 ‘Fort et al. (2020b) promoted the use cytoplasmic complete genome’ -> use of

Experimental design

see above

Validity of the findings

see above

Additional comments

see above

Reviewer 3 ·

Basic reporting

As reported by the title, the manuscript aims to identify the molecular diversity and taxonomic relationships of foliose Ulva species associated with green tides in France. However, the results are based on 3 sampling sites only and therefore the title “Molecular genetic diversity of the macromorphological taxon Ulva cf. rigida along French Atlantic coasts” is somewhat mis-leading.
A special focus has been placed on the species Ulva cf. rigida. The study describes the molecular genetic diversity of seaweeds morphologically related with Ulva rigida at three sites along the French Atlantic coast.

As the main finding of the study, the authors highlight the presence of five tufa haplotypes which were delineated into four species by phylogenetic techniques. The manuscript appears correctly structured and figures and tables present the main results. The text and language has improved but still needs improvement and I suggest to have it revised and checked by a native English speaker.

Experimental design

Although, the authors give reasons why sampling was conducted in winter/spring only, it appears to me that an additional sampling during vegetation-maximum (summer/early fall) would have largely benefited the study. With the present dataset the authors can verify the samples as “present at green tide locations” but that the species found are involved in green tides can in my eyes not be validated by the present study.

Although, the authors now state that they used the ASAP method for delimitation purposes it appears unclear to me on which basis the clusters are defined? What is the cut-off to delimit a cluster/species? Why is U. rigida and U. laetevirens defined as separate species but as common cluster?

Although, the authors state that tufA is the marker of choice to apply for species delimitation and identification in green algae the available datasets for rbcL contain more typus sequences. The dataset would still largely benefit from including rbcL as well.

Validity of the findings

One of my main points is the application of names to clades and species clusters. The authors correctly highlight clades which were identified with holotypes/Lectotypes, however every single cluster or species that is not validated by the molecular validation of a lectotype should be stated as such and identified by cf.. E.g. Ulva cf. linza etc.. For this see also the latest literature of Hughey et al. (as cited in your manuscript).

The conspecificity/ non-conspecificity of U. rigida and U. laetevirens should be explained more precisely and in more detail. Although, you emphasize that those are two different species I suggest to include the reasons why they are often regarded as one species (see also Hughey et al. 2020; cited in your manuscript).There are several more literature evidences which regard both species as conspecific, especially due to the biological ability to form fertile hybrids.

Additional comments

Line 150: Why ca.? Give exact number

Line 160: Name the ID of the holotype and provide herbarium code

Line 167 pp.: When focusing on the Jan/Feb period, and previously mentioning that green tides are more prevalent in the summer/fall time, how could you make sure the individuals taken at these sites really tend to proliferate?

Line 181: Please explain. Do you mean you disregarded this other criteria?

Line 231: U. compressa is not obligat tubular, neither is U. linza, nor U. intestinalis (monostromatic occurrences in Baltic Sea tend to form green tides [Bäck, Saara, Annamaija Lehvo, and Jaanika Blomster. "Mass occurrence of unattached Enteromorpha intestinalis on the Finnish Baltic Sea coast." Annales Botanici Fennici. Finnish Zoological and Botanical Publishing Board, 2000.]. Rephrase.

Line 264: “private” rephrase. Perhaps something like solely found at…

Line 283: Rephrase “our haplotypes”

Line 283 pp.: Since the manuscript aims to clarify taxonomic relations I would suggest to include cf. in the species names where no “clear” identification of the clade is possible. Thus, I suggest to use this consistently in all entity names which were not validated by genetic examination of the type (Holotype, Lectotype …).
Additionally I would rephrase the presentation of clades in the results. I agree, that clades which only contain sequences from one “species” can be named accordingly (see the cf. situation above), however perhaps include “identified as”.E.g.: “The first one contains 68 sequences identified as U. laetevirens”. This leaves it open for the discussion to discuss the correct taxonomic affiliation.

Line 283: Why do you delimit 4 and not 5 clades?

Line 312: Why Ulva specific? There is no evidence they do not align with other green algae as well?

Figure 1: The first graph of France looks warped. Adjust the CRS layer. (If done in QGIS or GIS you can use the function”set layer from CRS” by right clicking on the map)

Figure 2: Caption  “…gene chloroplastic gene.” Delete one “gene”

In general it is hard to really identify the clades on names and I would adjust this (except for the clades containing a holotype sequence) by adding a cf.. For sure, there will be changes in the future when more type material gets sequences, thus it enables for keeping track.

---

## Round 0.3 · Minor Revisions

Thank you for the thorough revisions to the paper. I believe they have addressed the referees' comments. I note that the abstracts on the covering page and main manuscript are different. Perhaps the first is an older version. The second is better. However, here and elsewhere please avoid isolated single parentheses [(1) not 1)]. Also, do not start sentences with abbreviations (e.g., lines 396, 477, 483). Please check throughout the manuscript for any editorial issues like this and resubmit.

---

## Round 0.4 · Minor Revisions

At your request, regarding your recent email about a recent paper revising Ulva genetics that impacts on this paper, we are sending the paper back to you to make the revisions you feel desirable.

---

## Round 0.5 · accepted · Accept

Thank you for the updates to the genetics and taxonomy.